# Correlates of Theta and Gamma Activity during Visuospatial Incidental/Intentional Encoding and Retrieval Indicate Differences in Processing in Young and Elderly Healthy Participants

**DOI:** 10.3390/brainsci14080786

**Published:** 2024-08-02

**Authors:** Mariana Lizeth Junco-Muñoz, Oliva Mejía-Rodríguez, José Miguel Cervantes-Alfaro, Adriana del Carmen Téllez-Anguiano, Miguel Ángel López-Vázquez, María Esther Olvera-Cortés

**Affiliations:** 1Laboratorio de Neurofisiología Clínica y Experimental, Centro de Investigación Biomédica de Michoacán, Instituto Mexicano del Seguro Social, Morelia 58060, Michoacán, Mexico; mariana.junco@umich.mx; 2Facultad de Psicología, Universidad Michoacana de San Nicolás de Hidalgo, Morelia 58194, Michoacán, Mexico; 3División de Investigación Clínica, Centro de Investigación Biomédica de Michoacán, Instituto Mexicano del Seguro Social, Morelia 58060, Michoacán, Mexico; olivamejia@yahoo.com; 4Laboratorio de Neurociencias, División de Estudios de Posgrado, Facultad de Ciencias Médicas “Dr. Ignacio Chávez”, Universidad Michoacana de San Nicolás de Hidalgo, Morelia 58194, Michoacán, Mexico; miguelcer43@hotmail.com; 5Division of Graduate Studies an Research, Tecnológico Nacional de México, Morelia 58120, Michoacán, Mexico; adriana.ta@morelia.tecnm.mx; 6Laboratorio de Neuroplasticidad, División de Neurociencias, Centro de Investigación Biomédica de Michoacán, Instituto Mexicano del Seguro Social, Morelia 58060, Michoacán, Mexico; migangelv@yahoo.com.mx

**Keywords:** aging, visuo-spatial learning, incidental learning, intentional learning, theta activity, gamma activity

## Abstract

Incidental visuospatial learning acquired under incidental conditions is more vulnerable to aging than in the intentional case. The theta and gamma correlates of the coding and retrieval of episodic memory change during aging. Based on the vulnerability of incidental coding to aging, different theta and gamma correlates could occur under the incidental versus intentional coding and retrieval of visuospatial information. Theta and gamma EEG was recorded from the frontotemporal regions, and incidental/intentional visuospatial learning was evaluated in young (25–60 years old) and elderly (60–85 years old) participants. The EEG recorded during encoding and retrieval was compared between incidental low-demand, incidental high-demand, and intentional conditions through an ANCOVA considering the patient’s gender, IQ, and years of schooling as covariates. Older adults exhibited worse performances, especially in place–object associations. After the intentional study, older participants showed a further increase in false-positive errors. Higher power at the theta and gamma bands was observed for frontotemporal derivations in older participants for both encoding and retrieval. Under retrieval, only young participants had lower power in terms of errors compared with correct responses. In conclusion, the different patterns of power and coherence support incidental and intentional visuospatial encoding and retrieval in young and elderly individuals. The correlates of power with behavior are sensitive to age and performance.

## 1. Introduction

Aging is characterized by a decline in several cognitive capabilities, with high variability in different cognitive domains. Behavior flexibility, executive function, declarative memory, and visuospatial learning and memory abilities are among the most vulnerable [1]. One of the factors that strongly influences the impact of aging on memory is the intentionality to acquire information. Compared with intentional encoding, incidental encoding, defined as the acquisition of information without an intended effort, is more vulnerable to aging [2,3,4]. In addition, discrete item information is preserved, but the association of items is impaired in aged people in terms of face–name (e.g., name–face associations were impaired compared with only names or only faces), word–color, and cue–context associations, among others [2,3,5]. Additionally, the combination of these attributes (incidental, visuospatial, and item association) results in greater difficulties in mature adults (45–65 years old) compared with young adults (18–40 years old) [6]. It has been proposed that incidental and intentional memory processes take place in the temporal lobe structures, possibly through different circuits. The temporal lobe is recognized as a crucial structure in the processing of declarative memory for both encoding and retrieval [4]. Neuroimaging studies have shown increased activity in the medial temporal lobe (MTL) during encoding in both intentional and incidental tasks for novel compared with familiar items [7,8]. Additionally, the subsequent effect of memory retrieval is observed in MTL activation during encoding, and increased activation was observed in fMRI for unfamiliar information compared with familiar information in the anterior temporal–medial cortex [9,10]. In addition, several studies have reported increased activity in the anterior temporal–medial region during successful retrieval, whereas no changes were found in other works [9,10,11,12]. It has been suggested that the activation of the anterior MTL occurs when relational information is being processed, whereas posterior activation takes place with familiarity processing [8,12,13]. Additionally, neuroimaging studies have shown the differential participation of the left inferior frontal cortex and the right dorsolateral prefrontal cortex, being more activated under intentional than incidental memory, whereas hippocampal activation was related to successful retrieval but not with the intentionality of remembering [4]. Thus, at present, it is not clear which networks are involved with respect to the intentionality of memory. Declarative memory has been related to changes in power and coherence in the theta and gamma bands expressed in the frontotemporal circuits [14,15,16]. Early works by Klimesch and coworkers showed an evoked increase in theta power associated with the successful encoding of memory in the temporal areas [17], as well as an induced reduction in theta power through the encoding process, which was also observed in the frontal areas [18]. In addition, theta synchronization between the frontal and parietal–temporal posterior regions has been observed during success memory encoding [19]. A subsequent memory effect on EEG oscillations with increased theta and reduced alpha evoked power has been observed under the encoding of semantic memory. Moreover, a retrieval-related increase in theta power and theta–gamma expression during successful memory encoding has been reported [18,20,21]. With regard to incidental associative memory, increased theta power in the left frontal areas during successful word–word and word–color association encoding was observed. In addition, the coherence values were higher at the theta frequency in the right and left frontal–posterior areas when participants correctly associated word–color information [22]. As with memory, the activity of the brain changes with age. The main changes in EEG activity with aging reported under rest include a general reduction in power but a global increase in the theta/alpha ratio, possibly due to a general increase in slow-frequency (delta and theta) activity [23,24]. Thus, changes in EEG activity related to memory performance coexist with changes related to normal aging and with pathology-related changes. This increases the difficulty in elucidating whether changes in EEG are a reflection of different but efficient modes of memory processing or indicative of a pathological aging process. Lopez-Loeza et al. [6] evaluated the incidental visuospatial memory performance of young and mature adult participants (18–25 years old and 45–65 years old, respectively) and observed a reduced span but similar efficiency in mature adults under incidental conditions. Meanwhile, under intentional conditions, the span was similar, but mature adults exhibited reduced accuracy in placing the objects into the observed positions. The authors observed increased theta activity in the frontotemporal areas in mature adults compared with young participants. When the authors grouped the participants by performance efficiency and compared their theta activity, the less efficient group showed higher frontal theta and gamma activity compared with the efficient group. The main age-related difference in EEG activity was observed at gamma frequencies, whereas theta and gamma were principally related to the efficiency at frontal derivations. From the above, we hypothesized that older participants would be deficient in incidental/intentional visuospatial coding and retrieval, and these deficiencies would be related to differences in the theta and gamma patterns recorded during the test. To test this hypothesis, in the present work, visuospatial incidental/intentional learning and memory were evaluated and compared between young and old participants. Additionally, an incidental high-demand condition was tested to determine whether incidental visuospatial memory showed a marked decline in healthy participants. Moreover, the theta and gamma activity were analyzed to assess whether performance and age interact during incidental and intentional processing and between encoding and retrieval.

## 2. Materials and Methods

Forty-nine participants from two age groups were included in this study: one young group (28 participants, 11 women, 25–45 years old) and one elderly group (20 participants, 13 women, 60–87 years old). The participants were invited from the “Unidad de Medicina Familiar # 83” of the “Instituto Mexicano del Seguro Social”, and informed written consent was provided by each participant. All procedures were performed in accordance with the National Ethics and Research Committee of the Instituto Mexicano del Seguro Social (CNCI: R-2014-785-058) and in accordance with the Declaration of Helsinki.

### 2.1. Inclusion Criteria

All participants had normal or corrected visual acuity. Beck’s depression inventory was applied to young participants and the Yesavage inventory to elderly participants [25]. The Montreal Cognitive Assessment (MoCA), adapted to the Mexican population [26], was applied to all participants. In addition, the PC version of the Raven test and the inverted figures and squares WAIS subscales were applied; the latter served to obtain the intelligence quotient (normal low, normal high and superior) and perceptual reasoning index, respectively [27]. The Raven test for PC also enabled the elderly participants to become familiar with the use of the laptop and mouse. For older adults who could not manipulate the mouse due to physical conditions, a facilitator was responsible for pressing the corresponding button, and the participant carried out a movement similar to the one that would be performed when using a mouse. Additional demographic data were obtained, including the participants’ years of schooling, general health state, and use of prescription drugs.

### 2.2. Exclusion Criteria

Participants with a history of cerebral damage, the use of prescribed drugs with action on the central nervous system, hypertensive diseases, and/or uncontrolled metabolic diseases were excluded from the study. Participants with scores of less than 18 on the MoCA Test Spanish 7.1BLIND, less than 50 in the Raven test, higher than 11 on Beck’s inventory, and higher than 10 on Yesavage’s inventory were excluded from this study. Finally, 3 participants who were unable to control their movementsthus affecting the EEG recordings (tics, jaw movements, head movements) were excluded.

### 2.3. Incidental/Intentional Visuospatial Task

The task was performed in an isolated room where the participants sat in front of a laptop monitor (70 cm distance to the eyes). Once the electrodes were placed, instructions for the procedure were given to the participants. The visuospatial task was realized based on that reported by Lopez-Loeza et al. [6], with some modifications, using the software developed by Tellez-Anguiano et al. [28].

The task started with the instruction to observe a fixation central cross on the screen for 60 s (baseline condition), and then a 60-s recording was taken with the eyes closed. Then, the image of a maze with seven objects embedded in it and two objects at the start and end points of the correct pathway was presented to the participants (Figure 1), with the instruction to “mentally solve the maze and indicate when finished”. Once the participant had solved the maze, the instruction was given to trace the pathway (using the mouse; data not analyzed), followed by the instruction to observe the maze (without any other insight) for 30 s (corresponding to the incidental low-demand encoding condition). After this, an image of one object (one that was in the maze or one new object) appeared in the center of the screen for two seconds, followed by the question, was this object in the maze? If the participant answered “no” (by clicking with the mouse), another object was presented on the screen for 2 s, and the question was presented again. If the participant answered “Yes”, the image of the maze without objects appeared on the screen, and the participant had to drag the item to the position remembered using the mouse. This part of the task was completed with the presentation of 8 objects embedded in the maze, interspersed with 7 new objects. The time at which the participants responded to each item was registered by the software and used to mark a point, from which an EEG sample of 4 s was analyzed. This period was considered as the incidental low-demand retrieval condition regarding the 8 objects that were present in the maze. In addition, the two seconds during which a new object was observed were considered and analyzed as the incidental high-demand encoding condition, because the participants had a shorter period of time (two seconds) to observe each object than in the incidental low-demand encoding. In addition, the participants were required to determine whether the item was new or familiar at that time, and they were not notified that this information would be asked for later. Once the 8 objects had merged in the maze and the 7 interspersed objects were observed, the maze with the eight objects was presented again to the participants, with the instruction to study the objects’ positions (for a 30-s period). This was considered the intentional encoding condition. Once the study time had elapsed, an object appeared on the screen for 2 s, followed by the question, was this object in the maze? If the participant answered “Yes”, the image of the maze was presented on the screen, and the participant had to drag the cursor to the position remembered. The period of 4 s (for each of the 8 studied objects) before the time of the response was analyzed and considered the intentional retrieval condition. If the participant answered “No”, another question was presented—did you see this object before, or is it a new object? Thus, the participants had to discriminate between the seven objects previously presented as new in the first incidental test and 7 new objects that were interspersed. The period of 4 s before the participant’s response was analyzed as the incidental high-demand retrieval condition. This part continued until the participant responded to the 7 objects used for contrast in the first test (incidental low) and the seven new objects interspersed with the 8 studied objects. Finally, a baseline open-eye EEG was recorded for 60 s; this was the baseline record included in this study, because the first record was obtained when the elderly participants were highly anxious compared with the young participants.

The responses of the participants were stored by the software and analyzed to obtain the number of false-positive errors (when the participant responded “yes” for objects that were not in the maze or were not new), false-negative errors (when the participant answered “no” for objects that were in the maze or were new), and total errors (the two types summed) for each behavioral subtest (incidental low demand, incidental high demand, and intentional). In addition, the number of object errors (objects that were not in the maze but were placed in it by the participants), place errors (objects that were or were not in the maze but were placed in the incorrect position by the participants), and correct object–place associations (correct objects placed in the correct positions) were compared for the incidental low-demand and intentional conditions. All behavioral variables described were compared between the groups with the Mann–Whitney U test, and intragroup comparisons were performed to assess the differences between the incidental and intentional acquisition of the information at each age through Wilcoxon’s paired test.

### 2.4. EEG Records

EEG records for the frontotemporal areas (Fp1, Fp2, F3, F4, F7, F8, T3, T4, T5, and T6), according to the 10–20 international system, referring to short-circuited and grounded ear lobes, were obtained. Electrodes to register blinks and cardiac pulse potentials were also placed. The EEG signals were amplified (Neuro-Data Acquisition System Mod. 15 Grass for 36 channels). The records were acquired and stored with Gamma software for Mod. 15 (Grass telefactor), with the filters set at 1–300 Hz and a sample frequency of 1024 Hz, and stored on a PC to be analyzed offline. EEG recording started simultaneously with the memory test, and the test software marked the time of each behavioral condition, the time for which each object was displayed, and the responses of the participants [28].

The encoding and retrieval EEG records were sampled and analyzed separately, each divided into behavioral conditions as described in the behavioral task section. Baseline records were taken and compared with the encoding and retrieval stages, including incidental low-demand, incidental high-demand, and intentional. Moreover, correct and incorrect responses were compared only for retrieval, as the encoding process was obtained from a continuous record. The length of the records depended on the behavioral condition as described above: 30 s for incidental low-demand encoding and intentional encoding; 4 s for incidental low-demand, incidental high-demand, and intentional retrieval; and two seconds for incidental high-demand encoding.

The raw EEG records were visually inspected, and artifacts were eliminated using the eeglab12_0_1_0b utility from MATLAB. Then, samples corresponding to each behavioral condition were obtained and subjected to a Fourier Transform (samples of one second with 50% overlapped windows, using the periodogram method with MATLAB scripts) to obtain the mean spectrum power of the theta (4–8 Hz) and gamma (low gamma, 30–50 Hz) bands. Coherence values were computed as the magnitude square coherence using Welch’s averaged periodogram method. Normalized power (natural logarithm) and coherence values were obtained in bins of 1 Hz of the theta and gamma bands and were compared between groups and conditions with an ANCOVA, with the participants’ gender, intellectual quotient (IQ), and years of schooling as covariates. Partial correlation coefficients were obtained between the behavioral variables and their corresponding EEG power values, considering gender as an independent partial factor and the IQ and years of study as dependent partial factors. When significant correlations were found (*p* < 0.01), the analysis was repeated, including the age as a partial correlate. If the initial significant correlation was lost, we considered the correlation to be age-dependent; when the correlation value did not change after age adjustment, we considered both age and efficiency to be significantly correlated with the EEG power.

## 3. Results

Descriptive data from the samples are shown in Table 1. Younger participants had significantly higher IQ values and more years of schooling than the elderly group. The young participants presented values in the normal high and superior ranges (2 and 3), whereas the older group presented normal low and high values (1 and 2). In addition, the elderly group had lower values in the MoCA test, whereas no differences were observed in the WAIS test.

### 3.1. Incidental/Intentional Visuospatial Test

In the incidental low-demand condition, no differences were observed in the number of correct responses (U_1df_ = 256.500, *p* = 0.103), total errors (U_1df_ = 200.500, *p* = 0.090), and false-negative (U_1df_ = 221.000, *p* = 0.207) and false-positive (U_1df_ = 236.500, *p* = 0.161) errors. In the comparison of place–object associations, the elderly participants obtained fewer correct responses (U_1df_ = 379.5, *p* = 0.035, d_Cohen_ = 0.630), more place recall errors (U_1df_ = 167.5, *p* = 0.014, d_Cohen_ = 0.722), and more false-positive errors (U_1df_ = 218.500, *p* = 0.025, d_Cohen_ = 0.378) than the younger group. No differences in object errors were observed (Figure 2).

In the incidental high-demand condition, the elderly group exhibited more total (U_1df_ = 124.000, *p* = 0.001, d_Cohen_ = 1.067), false-negative (U_1df_ = 187.000, *p* = 0.045, d_Cohen_ = 0.585), and false-positive (U_1df_ = 140.000, *p* = 0.001, d_Cohen_ = 0.933) errors and had a smaller number of correct responses (U_1df_ = 425.500, *p* = 0.002, d_Cohen_ = 0.978) than the younger group.

In the intentional test, the elderly group exhibited more total (U_1df_ = 174.501, *p* = 0.016, d_Cohen_ = 0.672) and false-positive (U_1df_ = 186.500, *p* = 0.023, d_Cohen_ = 0.588) errors than the younger group and had a smaller number of correct responses (U_1df_ = 385.500, *p* = 0.016, d_Cohen_ = 0.672), whereas no difference in false-negative errors was observed (U_1df_ = 246.500, *p* = 0.341). In the place–object associations, the elderly participants exhibited more place (U_1df_ = 177.500, *p* = 0.016 d_Cohen_ = 0.651, Figure 2) and false-positive errors (U_1df_ = 138.000, *p* < 0.001, d_Cohen_ = 0.949) and had a smaller number of correct responses (U_1df_ = 377.500, *p* = 0.035, d_Cohen_ = 0.616). No differences were observed in the false-negative errors and object errors (U_1df_ = 246.000, *p* = 0.334 and U_1df_ = 296.000, *p* = 0.675, respectively). In summary, the elderly participants were less efficient in terms of their incidental low-demand (for place–object associations), incidental high-demand, and intentional performance on the visuospatial task.

To assess whether the young and elderly participants showed a similar increase in performance after the intentional study of the items, an intragroup comparison of the scores obtained between the incidental (low-demand) and intentional conditions was performed. The number of correct responses increased under the intentional compared with the incidental low-demand test for both young (z = 4.655, *p* < 0.001, d_Cohen_ = 3.7) and elderly (z = 3.831, *p* < 0.001, d_Cohen_ = 3.321) participants. The young (z = −2.591, *p* = 0.010, d_Cohen_ = 1.123) but not the elderly participants (z = −1.792, *p* = 0.073, d_Cohen_ = 0.875, whereas both groups had a reduced number of false-negative errors (z = −3.256, *p* = 0.001, d_Cohen_ = 1.561; z = −2.454, *p* = 0.014, d_Cohen_ = 1.313). No changes between the incidental and intentional numbers of false-positive errors occurred in the younger participants, but an increase was observed in the elderly participants (z = 2.069, *p* = 0.039, d_Cohen_ = 1.043) (Figure 3). Thus, the elderly group was unable to further reduce its total errors after the intentional study of the positions due to the increase in the number of false-positive errors. Finally, the learning of place–object associations increased under intentional effort for young (z = 3.939, *p* < 0.001, d_Cohen_ = 2.229) and elderly participants (z = 3.587, *p* < 0.001, d_Cohen_ = 2.686). Additionally, the young participants, but not the elderly ones, showed reduced object errors (z = −2.428, *p* = 0.015, d_Cohen_ = 1.032) after the intentional study (Figure 3).

### 3.2. EEG Activity

#### 3.2.1. Theta and Gamma Activity during Encoding

Significant differences were observed in the mean power for all derivations recorded for both the theta and gamma bands, in which the elderly participants showed higher power values than the younger participants (Table 2, main effect). Additionally, the interaction of the group and behavioral condition was significant for all derivations and for the two bands (interaction statistic results, Table 3).

The paired comparisons of the theta band showed a general pattern in which the EEG power of the young participants had lower values under the incidental low-demand, incidental high-demand, and intentional conditions compared with the baseline record; meanwhile, the elderly participants had similar power values throughout all behavioral conditions. In addition, the EEG power of the young participants was lower than the power of the elderly participants in all test conditions, with the exception of the baseline record.

In the incidental high-demand condition, a significant and negative correlation occurred between the correct responses and the T6 power, but this correlation was lost after age adjustment, indicating a strong age relation. Finally, no significant correlation between power and behavior in the intentional encoding was observed for the theta band (Table 4). In summary, more power in the temporal (T3, T4, T6) and frontal (Fp1, Fp2) derivations occurred in less efficient participants, but this was more strongly related to aging than efficiency. Additionally, significant correlations were observed only for incidental and not for intentional encoding.

In the gamma band, under incidental low-demand encoding, the greater power in T3, T4, and F4 was related to the minor number of correct responses and correct place–object associations. However, the positive correlations between the false-negative errors and T3 power and between the place–object false-negative errors and T4 power were significant. All correlations lost their significance after age adjustment, and the r values were significantly reduced (Table 4). In the incidental high-demand condition, the greater power in Fp1, Fp2, and F3 was related to the minor number of correct responses, whereas the greater power in Fp1, Fp2, and T6 was related to the greater number of false-positive errors. All correlations were lost after age adjustment, and the reduction in the r values was significant. However, positive correlations between the F4, T3, and T4 power and the number of correct responses, as well as negative correlations between the F4 and T3 power and false-positive errors, were observed after age adjustment. Thus, in the gamma band, higher power at the frontal (Fp1, Fp2, F3) regions was related to a smaller number of correct responses and was also dependent on age, whereas the power at F4, T3, and T4 initially showed a negative correlation with correct responses and with false-positive errors but showed a positive correlation after age adjustment, probably due to the performance dependence. Under intentional conditions, only the greater power in T3 and T6 was related to more place errors, but this was dependent on age (Table 4).

#### 3.2.2. Theta and Gamma Activity at Retrieval

##### Theta and Gamma Power

The EEG power from the retrieval periods was analyzed in each behavioral condition and separated into correct and incorrect responses. The analysis revealed a significant group effect for each derivation, similar to that observed in the encoding condition; the power was higher in the older group in both the theta and gamma bands (Table 2). In addition, a significant effect of the interaction of the group and behavioral condition was observed for each derivation in the two bands (Table 3). Whereas the theta power of the elderly participants did not change in magnitude throughout the different behavioral conditions or between correct and incorrect responses, the EEG power of the younger participants was lower for the retrieval of incorrect responses compared with correct responses and with the incorrect responses of the elderly participants. Specifically, the theta power recorded under the incidental low-demand condition was significantly lower for error responses than for correct responses in F3, F8, T3, T4, and T6. Meanwhile, in the incidental high-demand condition, the theta power was lower for all recorded derivations, with the exception of Fp2. Finally, in the intentional retrieval, a similar pattern was observed for all derivations (Figure 4).

In the gamma band power recorded during retrieval, reduced gamma power occurred in regard to errors compared with correct responses and the baseline record, in all derivations and tasks, in the younger but not in the older group. Additionally, as a result of the power reduction in the error responses in the young participants, the power was significantly higher for the older group, whereas, regarding the correct responses, no differences between the groups occurred (Figure 5).

##### Power Behavior Correlations

In the incidental low-demand condition, the correlations were significant and positive for the false-positive errors and T4 theta power; however, after age adjustment, positive and significant correlations were also observed for the F8 and T3 power. Additionally, significant and positive correlations were observed for Fp1, F3, F4, F7, F8, T3, and T4 with place errors. After age adjustment, the F3, F7, F8, and T4 correlations were significantly reduced, although those of F4 and T3 remained significant, indicating the strong effect of the performance on the theta power modulation. No significant correlations were observed between the power in incidental high-demand retrieval EEG and behavior. Finally, in the intentional retrieval condition, there were significant and negative correlations between the power in Fp2, F3, F7, T3, and T5 and the number of correct place–object association responses, all of which significantly changed after age adjustment and lacked significance, indicating strong modulation by age (Table 5).

In the gamma band, the power–behavior correlations were significant only under the incidental low-demand condition, in which the number of false-positive errors was higher when the power was also higher in the F7 and F8 derivations after age adjustment, indicating a performance relation between power and behavior. Additionally, positive correlations between the gamma power and the number of place errors in Fp2, F4, F7, and F8 were observed, and these all remained after the age adjustment, with the exception of that of Fp2, which was significantly reduced. Thus, all were better related to the efficiency of processing, with the exception of Fp2, which could have been related to aging. Finally, a negative correlation between the power in F3 and F4 and correct place–object responses, which remained after age adjustment, was observed (Table 5).

## 4. Discussion

The presented results are in line with the proposed increased vulnerability of incidental, relational, and visuospatial information with aging [2,4,6,29,30]. Our results show a reduced ability to perform place–object associations with aging under incidental low-demand conditions, which was possibly due to the increase in place and false-positive errors in the elderly participants. Meanwhile, under the intentional condition, only the place–object false-positive errors were related to age. Koster et al. [31] compared the incidental versus intentional acquisition of memory for objects in young adults and two groups of children (mean of 7 and 10 years old). The authors observed that all groups had better performance under the intentional compared with the incidental condition in terms of the features of the objects observed; thus, an increased number of false alarms under intentional conditions appears to be a hallmark sign of aging. Although they did not observe an increased rate of false alarms/hits at these ages under the intentional compared with the incidental condition, previous works have reported an increased rate of false alarms [2,32]. This increase in false alarms was considered a deficit in recollection, combined with the overexpression of familiarity, possibly as a result of the lack of distinction between the newly learned information and the memory resources, leading to increased familiarity. Moreover, this deficit could be related to reduced hippocampal-entorhinal functional connectivity, based on the suggested role of the hippocampus in fast episodic memory encoding and recollection. Meanwhile, the hippocampus-associated cortex encodes familiarity and is less affected by aging [33,34,35].

These results did not show differences with regard to incidental versus intentional encoding for either the theta or gamma power. Similarly, in the study by Lopez-Loeza et al. [6], which compared younger and mature adults, the theta power of young participants did not show differences between incidental and intentional types of encoding. The authors showed that mature adults had increased theta power compared with the baseline records for incidental memory and intentional learning at frontal recording sites and only for intentional learning at temporoparietal sites. However, when the authors grouped the participants by accuracy (high efficiency and low efficiency), the less efficient group showed minor theta and gamma power at baseline and in incidental conditions, which increased in intentional conditions at the same derivations. Meanwhile, the efficient group did not show changes throughout the conditions (they started with high baseline power, and it remained high throughout all conditions). In this study, minor theta and gamma power in the incidental low-demand, incidental high-demand, and intentional conditions compared with the baseline records was observed for both encoding and retrieval errors, only for young participants. However, Koster et al. [31], reported an increase in the theta power of intentional compared with incidental encoding at the frontal and parietal electrodes. However, in Koster’s work, the theta power was measured as the phase-locked activity evoked by the stimulus. Additionally, different studies have shown an increase in the power of theta when comparing subsequent remembered and subsequent forgotten items, with the activity recorded during encoding and retrieval [17,22,36]. However, reductions in theta power have also been observed in other studies. Specifically, intracranial records in epileptic patients showed decreased high-frequency theta activity with cognitive processing, whereas increases in power were observed at lower frequencies in the so-called slow theta range (<4 Hz) [37]. In addition, early works by Klimesch et al. showed reduced power in the theta band, related to better performance in a memory task for word lists [24]. In this work, the authors evaluated the tonic changes in EEG through a continuous test in four-second epochs (similar to our study), and the results were explained as changes in induced power, which could reflect the default network state and were modulated by the stimuli but not phase-locked with them [17,24,38,39]. In another study, Nicolas et al. [38] reported a reduction in theta activity in the fronto-central regions during the retrieval of episodic information, as evaluated after weeks or months of encoding. Moreover, a theta power decrease during encoding compared with pre-stimulus in young participants was observed by Rondina et al. [39], who also reported increased theta power between encoding (the first phase of the study) and retrieval (test phase), despite a general theta power decrease from the baseline to the task. However, the theta range analyzed in Rondina’s study was 2–7 Hz. In the present work, the elderly participants had low efficiency and greater theta power both during encoding and retrieval, whereas the young participants had a reduction in theta power compared with the baseline records under the encoding and retrieval of incidental or intentional memory, but they showed a greater reduction when errors were made, in line with the mentioned studies. In addition, during retrieval, the theta power mirrored the accuracy of execution only for the young participants (higher power occurred for correct responses compared with errors). In the present work, the epochs obtained during encoding were taken from continuous recordings without a specific signal of the processing of information (when the participants observed the maze without other instructions). Thus, the changes in the tonic power measured during an extended period in which encoding occurred are quite similar to those reported by Klimesch et al. as the induced power, whereas the retrieval occurred after the presentation of an object and prior to the emission of a response in a period of four seconds in which the processing and decision-making were performed. This is in line with the assumption that increased theta power is observed as an event-related response due to a mechanism designed to increase the noise/signal ratio, reflecting the selectivity of information processing [17,18].

Under encoding, increased theta activity and power was correlated with the behavioral variables only for the incidental conditions in the frontotemporal areas, but the significance was lost after age adjustment. Under retrieval, higher theta power occurred in the left frontotemporal areas, which was related to aging, whereas, under incidental low-demand conditions, the theta power was more closely related to the efficiency than the age. High power at F8, T3, T4, and F4 was related to the greater number of errors after age adjustment; thus, the theta power at these derivations was related to both aging and the accuracy of processing under incidental conditions. These results are in line with the hypothesis of the overactivation or recruitment of additional resources to achieve a similar function in elderly adults compared with young adults [40,41], and with the reported increase in theta activity related to aging [23]. Several studies have associated cognitive interference with theta activity expression, principally through the performance of interference paradigms, in which the prefrontal theta expression reflects the interference level [42]. Additionally, in the context of episodic memory, increased theta activity has been related to higher levels of interference during memory retrieval in healthy young humans [43]. In this sense, the impaired inhibition of interferent information with aging is proposed in inhibitory deficit theory [1], which states that deficits in attention, language, and memory with aging are due to a decline in inhibitory function, producing difficulties in the suppression of irrelevant information or competing representations [42,44,45]. In this study, the higher levels of theta activity observed in the older participants could be related to higher levels of interference from information irrelevant to the task during encoding, whereas, during retrieval, additional interference from other memory representations could occur. In accordance with this, the successful inhibition of interference was associated with a reduction in theta activity in the frontoparietal areas in young humans’ EEG [43]. This is in line with the reduced theta power recorded in the young participants in memory-related conditions compared with their own baselines and with the theta power of older participants, under both encoding and retrieval, in this study. Moreover, the higher theta activity seen during correct responses compared with errors in young participants could be related to theta memory processing. Thus, these results support the dual modulation of the theta power by age (with higher interference) and memory processing, and they highlight the importance of dissociating the two effects in the study of the EEG correlates of memory processes.

The expression of gamma power during encoding was more strongly related to age than performance (the correlations observed under incidental low-load and intentional conditions were lost after age adjustment); however, the gamma power was also related to efficiency. The expression of gamma power could be related to the cortical representations of the stimuli, which, initially unfamiliar, are observed and compared with the previously observed items. In this regard, the oscillatory activity induced in the gamma band at frequencies higher than 20 Hz, observed after the presentation of an unfamiliar stimulus, has been related to the cortical representation of the semantic and perceptual characteristics of the stimuli, particularly with indirect memory (incidental) [20,46]. Additionally, it has been reported that the gamma activity decreases with the repetition of stimuli (e.g., faces), being higher for new stimuli than for familiar stimuli [46,47]. In accordance with this, under retrieval, the greater gamma power for Fp2, F4, F7, and F8 was related to the greater numbers of false-positive and place errors, and the lower power at F3 and F4 was related to the greater number of correct place–object responses. Additionally, this is in accordance with the increased gamma power during encoding but reduced during retrieval (only in young participants and possibly altered in elderly participants). It is also in line with the better performance observed only under incidental conditions in this study, supporting the relevance of gamma activity for incidental processing [20,46]. Moreover, increased gamma power under retrieval errors could be related to a deficient process, resulting in the perception of previously seen items as new items. With regard to the changes in gamma activity associated with aging, it was previously reported that mature adults (45–60 years old) show increased gamma power related to age and performance in the frontotemporoparietal areas compared with younger adults (18–40 years old) [6]. Similar results were observed by Barr et al. [48] when comparing working-memory-related gamma activity between younger (19–29 years old) and mature (30–60 years old) adults. The power increase was correlated with age but not with performance in an n-back test, in which the two groups had similar performance. These results extend the observed increase in gamma activity in mature adults (45–60 years old), reported by Lopez-Loeza et al. [6], to older adults at all sites recorded.

## 5. Conclusions

In conclusion, these results indicate that incidental and intentional visuospatial relational memory is more vulnerable to changes related to aging. The EEG power of the frontotemporal theta and gamma bands underlying this type of memory changes with aging as a general effect, but also changes in relation to task performance. The greater theta power in young participants might be related to inhibitory processes, whereas, in the elderly, it may be associated with compensatory mechanisms. In addition, it was observed that the EEG power was responsive to errors only in the young participants. Meanwhile, the gamma activity increases in the elderly participants could be related to increased familiarity and reduced recall. Changes in power are related to both aging and visuospatial processing; thus, it is important to separate each influence on EEG activity recorded in cognitive evaluations when studying aging.

## Figures and Tables

**Figure 1 brainsci-14-00786-f001:**
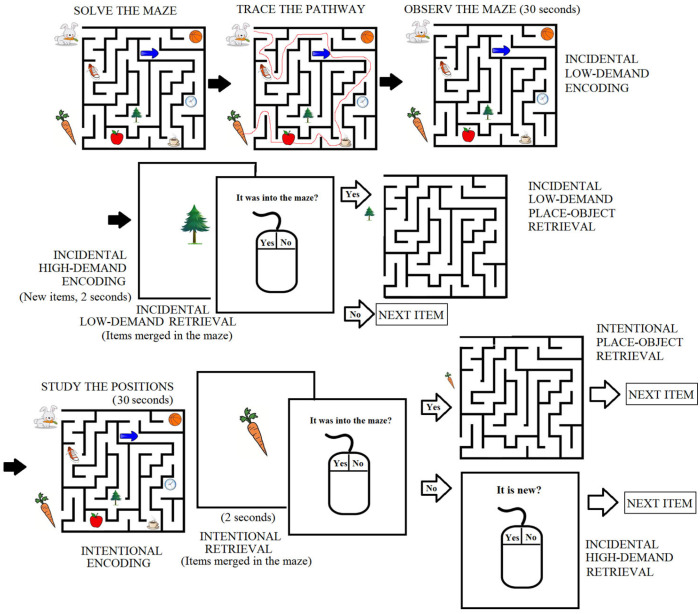
Diagrammatic representation of the incidental/intentional visuospatial test.

**Figure 2 brainsci-14-00786-f002:**
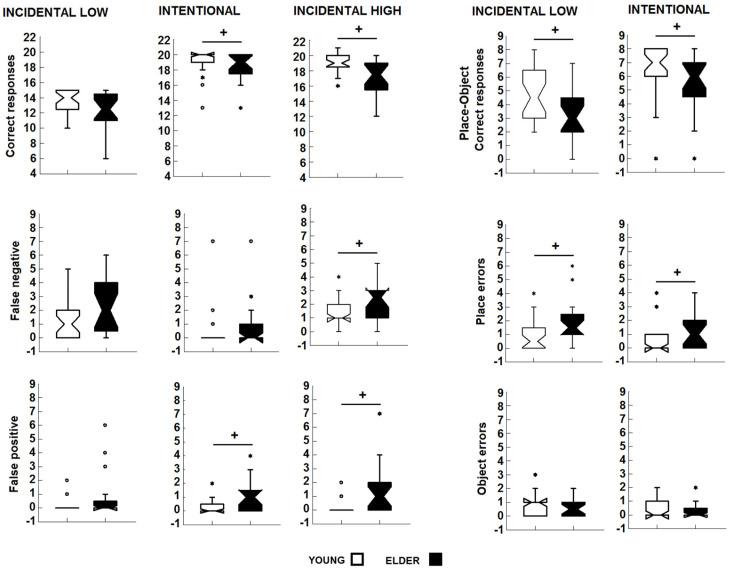
Box plots showing the intergroup comparisons of the scores obtained by the participants in the incidental low-demand, the incidental high-demand, and the intentional visuospatial memory test. +, young vs. elderly participants, other symbols ° and * are values out of range. Mann–Whitney U test: *p* < 0.05.

**Figure 3 brainsci-14-00786-f003:**
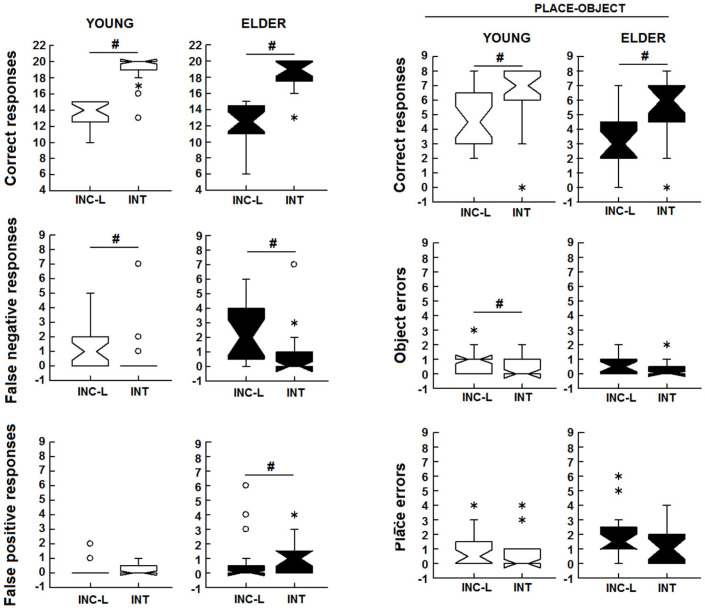
Box plots showing the intragroup comparisons of the scores obtained by the participants under the incidental low-demand and the intentional visuospatial conditions. #, incidental vs. intentional scores, other symbols ° and * are values out of range. Mann–Whitney U test: *p* < 0.05.

**Figure 4 brainsci-14-00786-f004:**
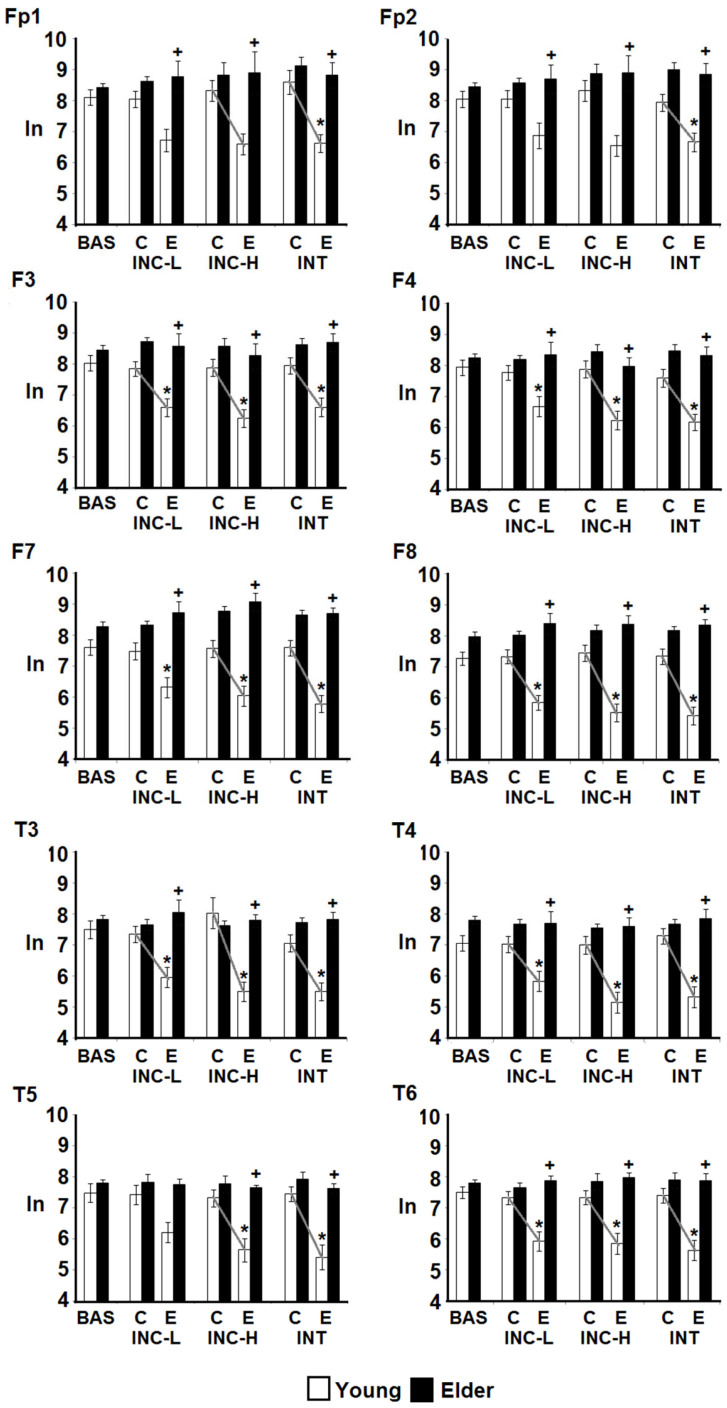
Intergroup and intragroup comparisons of the theta band EEG absolute power values (natural logarithm) obtained during retrieval in baseline (BAS), incidental low-demand (INC-L), incidental high-demand (INC-H), and intentional (INT) conditions. Comparisons between age, conditions, and responses (correct vs. errors C, E, respectively) are shown. Values are mean ± EEM. *, difference from baseline; +, difference between young and old participants; (line), difference between correct and error responses. *p* < 0.01.

**Figure 5 brainsci-14-00786-f005:**
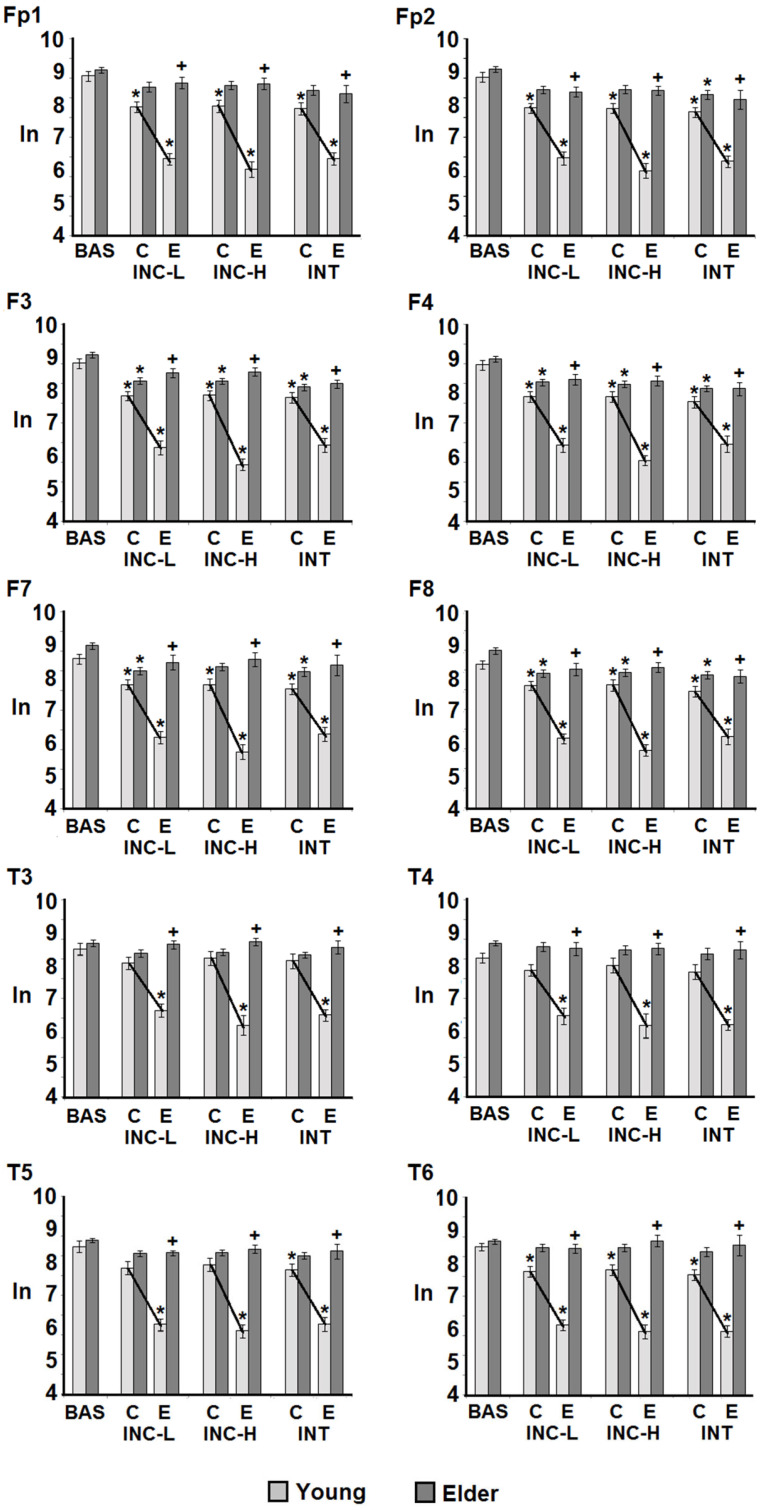
Intergroup and intragroup comparisons of the gamma band EEG absolute power values (natural logarithm) registered in retrieval, incidental low-demand (INC-L), incidental high-demand (INC-H), and intentional (INT) conditions. Comparisons between age and condition and between correct responses (C) and errors (E) are shown. Values are mean ± EEM. *, difference from baseline; +, difference between young and old participants; (line), difference between correct and error responses. *p* < 0.01.

**Table 1 brainsci-14-00786-t001:** Descriptive data of the participants.

Variable	Young (*n* = 28, 17 Women)	Elderly (*n* = 20, 13 Women)	*t*-Test/df	*p*
Age	33.321 (±6.608)	69.900 (±7.96)	−16.826, 36.1	<0.001
Years of schooling	15.500 (±3.097)	11.20 (±4.538)	3.671, 31.3	0.001
WAIS	100.143 (±11.906)	97.600 (±13.240)	0.684, 1	0.498
MoCA	19.24 (±1.261)	18.250 (±0.55)	4.527, 1	<0.001
			U test/df	
Raven	2.5 (1–3)	1.5 (1–3)	159, 1	0.006

For the age, years of schooling, WAIS, and MoCA results, which had a normal distribution, the mean ± standard deviation are presented, and the *t*-test and degrees of freedom (df) are reported. For the Raven category, the median and interquartile range are presented, and a paired comparison was performed with the Mann–Whitney U test.

**Table 2 brainsci-14-00786-t002:** Intergroup comparisons of EEG power of theta and gamma bands, recorded during encoding and retrieval.

	Encoding			Retrieval				
	Young	Elderly	F	d_Cohen_	Young	Elderly	F	d_Cohen_
Theta								
PF1	7.015 (0.173)	8.81 (0.139)	40.427	1.137	7.750 (0.141)	8.769 (0.111)	34.302	0.806
F3	6.861 (0.170)	8.643 (0.125)	40.024	1.132	7.410 (0.122)	8.498 (0.091)	49.943	0.973
F7	6.513 (0.185)	8.507 (0.100)	51.750	1.287	7.080 (0.123)	8.57 (0.073)	74.917	1.191
T3	6.195 (0.183)	7.904 (0.089)	44.854	1.198	6.895 (0.151)	7.765 (0.074)	19.853	0.613
T5	6.296 (0.199)	7.882 (0.070)	35.844	1.071	6.887 (0.136)	7.801 (0.089)	41.748	0.889
FP2	7.015 (0.180)	8.833 (0.140)	38.311	1.107	7.645 (0.131)	8.747 (0.098)	41.508	0.887
F4	6.853 (0.170)	8.541 (0.145)	32.063	1.013	7.34 (0.120)	8.326 (0.082)	37.646	0.844
F8	6.317 (0.156)	8.200 (0.108)	52.827	1.300	6.787 (0.119)	8.154 (0.071)	73.475	1.180
T4	5.921 (0.184)	7.887 (0.096)	45.437	1.200	6.563 (0.129)	7.706 (0.071)	37.163	0.839
T6	6.356 (0.153)	7.952 (0.069)	69.953	1.496	6.882 (0.115)	7.825 (0.080)	61.497	1.079
Gamma								
PF1	4.773 (0.215)	7.719 (0.124)	101.623	1.803	5.904 (0.183)	7.747 (0.094)	75.692	1.197
F3	4.700 (0.208)	7.365 (0.117)	88.201	1.650	5.708 (0.181)	7.430 (0.091)	63.758	1.099
F7	4.607 (0.213)	7.148 (0.119)	65.941	1.452	5.571 (0.178)	7.411 (0.100)	67.350	1.130
T3	5.013 (0.221)	7.393 (0.109)	63.023	1.420	6.109 (0.183)	7.512 (0.071)	37.027	0.873
T5	4.515 (0.218)	7.179 (0.113)	90.588	1.702	5.634 (0.187)	7.289 (0.068)	72.607	1.173
FP2	4.675 (0.221)	7.400 (0.122)	92.778	1.723	5.817 (0.179)	7.562 (0.095)	65.731	1.116
F4	4.734 (0.212)	7.091 (0.112)	52.654	1.298	5.706 (0.179)	7.247 (0.086)	44.488	0.918
F8	4.532 (0.209)	6.815 (0.117)	61.433	1.402	5.435 (0.170)	7.110 (0.090)	58.620	1.054
T4	4.862 (0.232)	7.482 (0.130)	49.279	1.256	5.724 (0.181)	7.551 (0.094)	43.710	0.910
T6	4.401 (0.203)	7.499 (0.146)	124.997	2.000	5.51 (0.174)	7.539 (0.080)	127.36	1.553

The means (SEM) of the power values are expressed as a natural logarithm, with the F and *p* values from the ANCOVA for group effects and with gender, IQ, and years of study as covariables; size effect (d_Cohen_) values are also shown (*p* < 0.001) for all comparisons.

**Table 3 brainsci-14-00786-t003:** Statistical results of the interaction by group and behavioral condition for encoding and retrieval regarding the theta and gamma band power.

	Encoding			Retrieval		
	F	*p*	d_Cohen_	F	*p*	d_Cohen_
Theta						
PF1	5.465	0.001	0.759	3.318	0.004	0.592
F3	5.459	0.001	0.759	4.875	<0.001	0.717
F7	4.730	0.004	0.707	6.007	<0.001	0.796
T3	5.789	0.001	0.782	5.233	<0.001	0.743
T5	3.974	0.010	0.648	3.544	0.002	0.612
FP2	4.871	0.003	0.717	3.126	0.006	0.574
F4	5.360	0.002	0.752	3.276	0.004	0.588
F8	5.048	0.002	0.730	8.260	<0.001	0.934
T4	4.016	0.009	0.651	4.699	<0.001	0.704
T6	8.485	<0.001	0.946	6.052	<0.001	0.799
Gamma						
PF1	15.810	<0.001	1.292	13.572	<0.001	1.197
F3	22.304	<0.001	1.534	19.619	<0.001	1.143
F7	17.060	<0.001	1.342	15.561	<0.001	1.281
T3	16.885	<0.001	1.335	13.298	<0.001	1.185
T5	17.031	<0.001	1.341	14.232	<0.001	1.226
FP2	12.216	<0.001	1.135	12.059	<0.001	1.128
F4	15.423	<0.001	1.276	13.847	<0.001	1.209
F8	14.950	<0.001	1.256	14.677	<0.001	1.245
T4	9.512	<0.001	1.002	7.524	<0.001	0.891
T6	15.445	<0.001	1.277	19.599	<0.001	1.438

The F and *p* values from the interaction between the group and behavioral condition were obtained in the ANCOVA with gender, CI, and years of study as covariables; the size effect (d_Cohen_) values are also shown.

**Table 4 brainsci-14-00786-t004:** Correlations between theta and gamma power and behavioral variables during the coding of information.

	Theta				Gamma			
Incidental Low-Demand								
	**Age-adjusted**				**Age-adjusted**			
		r, *p*	r, *p*	q_Cohen_		r, *p*	r, *p*	q_Cohen_
CR	T3 T4	−0.373, 0.032 −0.354, 0.034	**−0.257**, **0.097** **−0.254**, **0.099**	**0.129** **0.110**	T3 T4	−0.321, 0.033 −0.309, 0.034	**−0.191**, **0.109** **−0.213**, **0.102**	**0.139** **0.103**
FNE	Fp1 Fp2	0.350, 0.034 0.342, 0.036	**0.235**, **0.123** **0.238**, **0.119**	**0.126** **0.114**	T3	0.299, 0.048	**0.147**, **0.222**	**0.160**
P-O CR					F4 T3 T4	−0.289, 0.041 −0.376, 0.011 −0.304, 0.037	**−0.158**, **0.182** **−0.226**, **0.055** **−0.191**, **0.144**	**0.138** **0.165** **0.121**
P-O E P-O FNE					T3 T4	0.338, 0.024 0.294, 0.044	**0.170**, **0.154** **0.170**, **0.196**	**0.180** **0.131**
Incidental High-Demand								
CR FPE	T6	−0.438, 0.017	**0.003**, **0.981**	**0.473**	Fp1 Fp2 F3 F4 T3 T4 Fp1 Fp2 F4 T3 T6	−0.393,0.027 −0.409, 0.020 −0.364, 0.048 **−0.249**, **0.160** **−0.242**, **0.195** **−0.158**, **0.367** 0.369, 0.040 0.404, 0.021 **0.126**, **0.487** **0.070**, **0.713** 0.371, 0.042	**0.067**, **0.613** **0.057**, **0.652** **0.182**, **0.114** 0.257, 0.027 0.274, 0.032 0.292, 0.025 **0.023**, **0.862** **0.056**, **0.654** −0.264, 0.022 −0.337, 0.006 **0.007**, **0.954**	**0.482** **0.491** **0.566** **0.517** **0.528** **0.460** **0.410** **0.372** **0.397** **0.421** **0.397**
Intentional								
PE					T3 T6	0.332, 0.027 0.294, 0.047	**0.163**, **0.184** **0.105**, **0.348**	**0.181** **0.198**
Behavioral Variable								

Only significant correlations between the theta or gamma power and behavior are presented. Bold Pearson’s r and *p* values indicate correlations that changed after age adjustment (i.e., that were lacking or emerged). Bold Cohen’s q values indicate significant changes in r values. Cohen’s q indicates significant effect. Abbreviations: CR, correct responses; TE, total errors; FPE, false-positive error; P-O CR, place–object correct responses; PE, place errors.

**Table 5 brainsci-14-00786-t005:** Correlations between theta and gamma power and behavioral variables during the retrieval of information.

	Theta				Gamma			
Incidental Low-Demand								
	**Age-adjusted**				**Age-adjusted**			
		r, *p*	r, *p*	q_Cohen_		r, *p*	r, *p*	q_Cohen_
FPE	F8 T3 T4	**0.359**, **0.079** **0.384**, **0.059** 0.393, 0.044	0.335, 0.026 0.367, 0.042 0.381, 0.042	0.027 0.020 0.014	F7 F8	**0.257**, **0.087** **0.252**, **0.105**	0.236, 0.050 0.226, 0.029	0.022 0.028
PE	Fp1 F3 F4 F7 F8 T3 T4	0.415, 0.046 0.440, 0.023 0.474, 0.015 0.420, 0.042 0.401, 0.046 0.518, 0.007 0.429, 0.025	**0.355**, **0.087** **0.354**, **0.054** 0.403, 0.034 **0.300**, **0.099** **0.255**, **0.105** 0.419, 0.017 **0.362**, **0.055**	0.070 **0.102** 0.088 **0.138** **0.164** **0.127** 0.079	Fp2 F4 F7 F8	0.350, 0.038 0.280, 0.022 0.374, 0.007 0.372, 0.010	**0.255**, **0.072** 0.205, 0.036 0.293, 0.010 0.267, 0.007	**0.105** 0.080 0.091 0.117
P-O CR					F3 F4	−0.295, 0.033 −0.324, 0.025	−0.273, 0.051 −0.292, 0.046	0.024 0.035
Intentional								
P-O CR	Fp2 F3 F7 T3 T5	−0.338, 0.047 −0.356, 0.037 −0.358, 0.025 −0.396, 0.017 −0.336, 0.047	**−0.111**, **0.473** **−0.141**, **0.370** **−0.119**, **0.397** **−0.237**, **0.138** **−0.103**, **0.500**	**0.240** **0.230** **0.255** **0.177** **0.246**				
Behavioral Variable								

Bold r and *p* values indicate correlations that changed after age adjustment (i.e., that were lacking or emerged). Bold Cohen’s q values indicate significant changes in r values. Abbreviations: P-O CR, place–object correct responses; PE, place errors. No significant correlations between the gamma power and behavioral variables were observed for incidental high-demand or intentional conditions.

## Data Availability

The data presented in this study are available on request from the corresponding author due to to the authorization must be aproved by the Instituto Mexicano del Seguro Social.

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
