# Peer review of "Correlates of Theta and Gamma Activity during Visuospatial Incidental/Intentional Encoding and Retrieval Indicate Differences in Processing in Young and Elderly Healthy Participants"

_brainsci, 2024, doi:10.3390/brainsci14080786_

Round 1

Reviewer 1 Report

Comments and Suggestions for Authors

Dear Editor and Authors,

I reviewed the manuscript titled "Correlates of Theta and Gamma Activity During Visuospatial Incidental/Intentional Encoding and Retrieval Indicate Different Processing in Young and Elderly Healthy Participants." The study investigates visuospatial learning in young and elderly participants, using EEG data to identify psychophysiological biomarkers of incidental and intentional encoding and retrieval. Although the research question is intriguing, there are several methodological issues. The most significant are:

  1. The data and results are generally poorly detailed. It is unclear whether means or medians are reported, and whether the figures represent percentages or rates. Additionally, standard errors or other measures of variance are systematically ignored.
  2. The EEG data pre-processing relies on visual inspection, and the details of the data analysis methods are not reported. For example, parameters of the Fourier transformation and the specific theta and gamma frequency ranges are not provided.
  3. The task includes only a single trial per condition, which makes the results highly susceptible to individual variability.

Given these issues, I do not believe the study meets the journal's criteria.

Comments on the Quality of English Language

English language required extensive editing.

Author Response

  1. The data and results are generally poorly detailed. It is unclear whether means or medians are reported, and whether the figures represent percentages or rates. Additionally, standard errors or other measures of variance are systematically ignored.

Response: The authors apologize by the mistake. All tables have a foot indicating the values and statistics on the content, but when we charged the files, this footnote left mixed with the text. Al tables have now the respective footnote explaining the data reported and the statistics associated.

  1. The EEG data pre-processing relies on visual inspection, and the details of the data analysis methods are not reported. For example, parameters of the Fourier transformation and the specific theta and gamma frequency ranges are not provided.

Response: The parameters of Fourier and the ranges of theta and gamma band are now described in the text.

  1. The task includes only a single trial per condition, which makes the results highly susceptible to individual variability.

Response: Although the tasks can not be repeated by its incidental nature, the samples obtained in each part of the test are similar for each participant. In the incidental and intentional coding the samples from 30 seconds can be obtained; whereas in the retrieval the number of samples are 8 from the objects immersed in the maze (incidental low) or 7 (incidental high). The behavioral data were enough to show significant differences. In regard to the EEG samples, we used the Block design to intragroup comparisons. These ANOVA for blocks is the option to choose when the data show higher interindividual differences than intraindividual, that is the case of EEG power.

We consider that the variability is controlled by the normalization of the data (transformation into natural logarithm) and by the adequate statistical processing. In addition, the size effects were presented, and qChoen for changes in r values. All together allow us to consider that our data is solid.

  1. English language required extensive editing.

Response: the manuscript was revised by the editorial team suggested by the journal.

Reviewer 2 Report

Comments and Suggestions for Authors

Correlates of theta and gamma activity during visuospatial incidental/intentional encoding and retrieval indicate different processing in young and elderly healthy participants.

I found the manuscript potentially interesting, however, I have to write that the tables are not clear and not well-formatted in the main text. I have also tried to convert the PDF file into word, but the issue was not fixed. Unfortunately, this made the reading very difficult, as well as interpreting and understanding the results. Vg=i.e.?

My commentaries are as follows:

Introduction: the section is well written and clear. However, you must check the English for typos. I suggest modifying the “temporal medial lobe” with the medial temporal lobe, which is more common. Moreover, I suggest to be more specific in the hypothesis section. You described the study, but hypotheses are needed.

Methods- The sample, inclusion, and exclusion criteria are described in a good manner. Please, add the number of participants that were excluded, and the reason. I suggest to organize the description of the task in a better way. You need to add if the stages if the task were triggered with EEG system.

EEG: please, add more information about the EEG. It is not clear if you used a 32 or 64-channel system, as well as the software that was used to analyze the data.

Results: are interesting and the statistical analyses were conducted rigorously. Table 1 is ok. FNE is lacking in Table 4. However, the first raw needs to be fixed, since the p values are not completely clear, in particular adj p values. The same for Table 5. Moreover, the values in  BOLD are not clear. Indeed, the authors stated that the values in BOLD do not change after correction. This needs to be justified in a better way in the full text and mentioned in the table captions.

Discussion: In the first paragraphs you have summarized the results. I advise focusing on the principal objective of the studies (main problem- 1 sentence), and main results. Please, reduce this part, since could be quite redundant.

 “Specifically, intracranial records in epileptic patients showed decreased high-frequency theta activity with cognitive pro-cessing, probably related to the disengagement of the network default mode at theta fre-quencies (4-8 Hz), whereas increases in power were observed at lower frequencies in the so-called slow theta (< 4 Hz) [37]” This sentence is not clear and the reference to DMN is also unclear. You need to explain the role played by DMN in memory encoding and retrieval and its relation with Theta. 

Author Response

Introduction: the section is well written and clear. However, you must check the English for typos. I suggest modifying the “temporal medial lobe” with the medial temporal lobe, which is more common. Moreover, I suggest to be more specific in the hypothesis section. You described the study, but hypotheses are needed.

Response: the term “temporal medial lobe” was changed to medial temporal lobe, as the reviewer suggested, in addition, the hypothesis was included in the introduction section.

Methods- The sample, inclusion, and exclusion criteria are described in a good manner. Please, add the number of participants that were excluded, and the reason. I suggest to organize the description of the task in a better way. You need to add if the stages if the task were triggered with EEG system.

Response: The number of participants excluded and the reason was stated in the methods section, the EEG records were simultaneous and triggered by the behavioral software as is now indicated in the same section. The description of the task was corrected and we consider that is clearer now,  they can be found on pages 3 and 4 of the Word document.

EEG: please, add more information about the EEG. It is not clear if you used a 32 or 64-channel system, as well as the software that was used to analyze the data.

Response: The software for record and store of the EEG signals (Gamma, by Grass telefactor) is listed in the section, also the software to clean and section the EEG records (eeglab, matlab).

Results: are interesting and the statistical analyses were conducted rigorously. Table 1 is ok. 

FNE is lacking in Table 4. However, the first raw needs to be fixed, since the p values are not completely clear, in particular adj p values.

Response: The footnote was placed in the table 4, we apologize for the mistake, the footnote and table name were moved when the file template was filled.

The same for Table 5. Moreover, the values in  BOLD are not clear. Indeed, the authors stated that the values in BOLD do not change after correction. This needs to be justified in a better way in the full text and mentioned in the table captions.

Response: In fact, the values in bold changed after age adjustment, in addition the qCohen was obtained to test if the change in correlation was significant. This is now clearer in the results section and in the table captions.

Discussion: In the first paragraphs you have summarized the results. I advise focusing on the principal objective of the studies (main problem- 1 sentence), and main results. Please, reduce this part, since could be quite redundant.

Response: the first paragraph was reduced and is not redundant now.

 “Specifically, intracranial records in epileptic patients showed decreased high-frequency theta activity with cognitive pro-cessing, probably related to the disengagement of the network default mode at theta fre-quencies (4-8 Hz), whereas increases in power were observed at lower frequencies in the so-called slow theta (< 4 Hz) [37]”

This sentence is not clear and the reference to DMN is also unclear. You need to explain the role played by DMN in memory encoding and retrieval and its relation with Theta. 

Response: This comment: “probably related to the disengagement of the network default mode at theta frequencies (4-8 Hz)”, refers to the default mode brain state, more than the default network mode (our mistake). It refers to the reduction in theta activity observed from Close eyes state to open eyes state in cortical EEG, as an index of increased activation (Chen et al., NeuroImage 41 (2008) 561–574). However, as it is a weak argument and is disconnected of the rest of the discussion we decided to eliminate the phrase.

Round 2

Reviewer 1 Report

Comments and Suggestions for Authors

I do not have further comments.

Reviewer 2 Report

Comments and Suggestions for Authors

After reading the new version of the manuscript and the authors response, according to me, it has been improved.